# Evaluation of Three Serological Tests for Diagnosis of Canine Brucellosis

**DOI:** 10.3390/microorganisms11092162

**Published:** 2023-08-26

**Authors:** Fabrizia Perletta, Chiara Di Pancrazio, Diamante Rodomonti, Tiziana Di Febo, Mirella Luciani, Ivanka Marinova Krasteva, Marta Maggetti, Francesca Profeta, Romolo Salini, Fabrizio De Massis, Flavio Sacchini, Manuela Tittarelli

**Affiliations:** Istituto Zooprofilattico Sperimentale dell’Abruzzo e del Molise, 64100 Teramo, Italy; f.perletta@izs.it (F.P.); d.rodomonti@izs.it (D.R.); t.difebo@izs.it (T.D.F.); m.luciani@izs.it (M.L.); i.krasteva@izs.it (I.M.K.); m.maggetti@izs.it (M.M.); f.profeta@izs.it (F.P.); r.salini@izs.it (R.S.); f.demassis@izs.it (F.D.M.); f.sacchini@izs.it (F.S.); m.tittarelli@izs.it (M.T.)

**Keywords:** *Brucella canis*, diagnosis, serological methods

## Abstract

Canine brucellosis caused by *Brucella canis*, is an infectious disease affecting dogs and wild Canidae. Clinical diagnosis is challenging, and laboratory testing is crucial for a definitive diagnosis. Various serological methods have been described, but their accuracy is uncertain due to limited validation studies. The present study aimed to evaluate the performances of three serological tests for the diagnosis of *B. canis* in comparison with bacterial isolation (gold standard), in order to establish a protocol for the serological diagnosis of canine brucellosis. A panel of sera from naturally infected dogs (*n* = 61), from which *B. canis* was isolated, and uninfected dogs (*n* = 143), negative for *B. canis* isolation, were tested using microplate serum agglutination (mSAT), complement fixation performed using the *Brucella ovis* antigen (*B. ovis*-CFT), and a commercial immunofluorescence assay (IFAT). The sensitivity and specificity of the three serological methods were, respectively, the following: 96.7% (95% CI 88.8–98.7%) and 92.3 (95% CI 86.7–95.1%) for mSAT; 96.7% (95% CI 88.8–98.7%) and 96.5 (95% CI 92.1–98.2%) for *B. ovis*-CFT; 98.4% (95% CI 91.3–99.4%) and 99.3 (95% CI 96.2–99.8%) for IFAT. The use in of the three methods in parallel, combined with bacterial isolation and molecular methods, could improve the diagnosis of the infection in dogs.

## 1. Introduction

Canine brucellosis caused by *Brucella canis*, a rough species of the *Brucella* genus, is an infectious disease affecting dogs and wild Canidae [1]. The disease causes reproductive failures such as infertility or abortion in females, epididymitis in males, and high neonatal mortality [2]. Dog-to-dog transmission occurs during breeding or through oronasal contact with reproductive discharges following abortions. *B. canis* may also be shed with urine, feces, and nasal and ocular secretions. Pups may be infected in utero or perinatally [3]. These bacteria may be transmitted either by the venereal or oral route, more frequently infection occurs following contact with abortive material. In males, urine and seminal fluid represent an important source of infection [4]. Prolonged bacteraemia is a typical sign of canine brucellosis that persists from 6 months to 5 years [5]. Infected animals may also develop systemic symptoms, such as lymphadenitis, splenomegaly, diskospondilitis with possible neurological complications [3,6], or uveitis [7,8]. Frequently, infection remains subclinical [9] and favors disease spreading, especially when occurring in breeding kennels, and causing relevant economic losses [10]. The disease is of particular importance for dog breeders since infection with *B. canis* usually ends a dog’s reproductive career [2]. Evidences of *B. canis* infection have been recorded worldwide, in many territories of Central and South America, the Southern United States and Asia [11]. Observed seroprevalence ranged from moderate to high [12], especially among stray dog population [13,14]. Cases have also been reported in European countries, such as Germany [15,16], Hungary [17], Sweden [18], Switzerland [19], and the United Kingdom [20,21], further confirming the need for the implementation of disease surveillance activities in European countries [13,14,22,23,24,25,26,27]. Due to its zoonotic potential, canine brucellosis also represents a Public Health issue. Transmission to humans may occur by direct contact with infected dogs or their contaminated secretions, as well as through direct laboratory exposure. A limited number of human cases have been reported over the years [28,29,30], but the lack of specific diagnostic tests contributes to biased real data on disease occurrence, which could be underestimated [31]. Dog owners, personnel working in breeding kennels or with stray dogs, and laboratory workers handling infected material are more at risk of infection [32,33].

Conventional serological tests for brucellosis are based on lipopolysaccharide (LPS) obtained from smooth *Brucella* strains (reference strains *B. abortus* S99 and S1119-3, *B. melitensis* 16M) [34], and do not detect antibodies against *B. canis* [12] which is a rough strain lacking the lateral O-chain of the LPS, representing the major *Brucella* antigen against which the antibody response is directed [35,36]. Diagnostic antigens prepared from either homologous strains of *B. canis* or heterologous rough strains of the genus *Brucellae*, like *B. ovis*, have been proven to be effective for the detection of antibodies against *B. canis* [10]. The use of heterologous antigens relies on the high genetic and antigenic homology among *Brucella* species. Therefore, both *B. canis* and *B. ovis* strains could be used as an antigen with the same results [37]. The heterologous wall of antigens can cross-react completely, and could be used for the detection of antibodies against *B. canis* [38].

Identifying brucellosis-infected dogs is often challenging. A definitive diagnosis requires the culture of *B. canis* from the blood of infected dogs, a process that has relatively low sensitivity, is time consuming, and is somewhat technically impractical to apply to large populations of dogs. Several serological methods based on agglutination, immunoprecipitation, complement fixation, immune fluorescence, or ELISA have been described for routine testing [39,40,41,42]. Serological tests are valuable for screening large populations of dogs, such as breeding kennels, shelters or stray dog populations [25]. However, these tests suffer with uncertainties in the accuracy (false-positive and false-negative results may occur with these tests), due to the difficulties of carrying out validation studies, while isolation or detection of *B. canis* using bacteriological or molecular techniques are the most appropriate methods for disease confirmation [42]. Moreover, the combination of more than one laboratory test (direct and/or indirect), and repeated sampling of various biological specimens may be necessary for a conclusive diagnosis [11], representing a major challenge in laboratory investigations of canine brucellosis [43].

In the summer of 2020, *B. canis* infection was confirmed for the first time in a commercial breeding kennel in Italy [25], and outbreak management activities provided the opportunity to collect samples from a population of infected animals with the aim of defining the accuracy of laboratory testing for *B. canis* diagnosis.

The aim of this study was to validate a diagnostic protocol for serological diagnosis of *B. canis* based on the combination of three different serological tests. Two in-house laboratory procedures (microplate serum agglutination and complement fixation) and a commercial immunofluorescence assay were assessed using a panel of sera collected from *B. canis* naturally infected and uninfected dogs. Performances of individual and combined tests are presented and discussed.

## 2. Materials and Methods

### 2.1. Sera and Blood Collection

Sixty-one sera were collected from the infected animals of the 2020 Italian outbreak [25]. Uninfected dogs (*n* = 143) were also sampled from laboratory routine testing and selected from owned dogs using an anamnestic questionnaire to rule out canine brucellosis in these dogs. Both sets of sera were confirmed as positive or negative by EDTA whole blood culture for the detection of *Brucella* spp. Briefly, samples were analyzed using *Brucella*’s selective media as described in De Massis et al. (2021) [25], both solid and liquid for 4 weeks, and considered negative when no bacterial growth was observed at the end of the incubation period. Typical *Brucella* spp. colonies were tested by PCR for species identification [25].

### 2.2. Microplate Serum Agglutination Test (mSAT)

The mSAT test was carried out alongside a modified tube agglutination test described by Alton and colleagues [39], and the volumes were adapted to be performed in 96-well microplates [44]. This test primarily detects pentameric IgM antibodies, produced during the early stages of antibody response, and is characterized by a higher agglutination capacity compared to other isotypes [45].

*B. canis* antigen was prepared as previously described [39], with minor modifications. Briefly, *B. canis* strain RM6/66 (ATCC 23365) was grown on glycerol-dextrose agar or brain heart infusion agar roux in an aerobic atmosphere, at 37 °C for 48 h. *Brucella* colonies were collected from the surface of each roux, by washing with phosphate-buffered saline (PBS) containing 0.06% of formalin, and the resulting suspension was heat-inactivated at 70 ± 3 °C for 1 h. The inactivated suspension was washed three times in a refrigerated centrifuge at 10,000× *g* for 30 min to remove debris. The pellet was suspended in PBS containing 0.05% of formalin (37%) in a 4.5 *w*/*v*. The intermediate product is titrated in order to obtain a final product 10× concentrated, which has to be diluted with Tris-maleate buffer (Tmb) pH 9 ± 0.5 before use.

The mSAT analysis was performed in 96-well U-shaped microplates. Before testing, serum samples were diluted 1:10 in Tmb and then tested as two-fold dilutions by dispensing equal volumes (50 μL) of Tmb diluted serum and *B. canis* antigen. The testing dilutions of the serum ranged from 1:20 to 1:640. Positive and negative control sera were treated and analyzed as described for serum samples. *B. canis* rabbit hyperimmune serum was used as positive control (showing 100% agglutination at 1:160 dilution), and a negative serum from a *B. canis* naïve rabbit was used as negative control (showing no agglutination at 1:20 dilution). Rabbit hyperimmune serum was produced using *B. canis* RM66 strain to sensitize animals (authorization number 207/2021-PR). Plates were sealed and incubated at 37 °C for 48 h. Sera showing 100% agglutination at dilutions ≥1:20 were considered positive. Results were expressed as the highest serum dilution showing 100% agglutination (Figure 1).

### 2.3. Complement Fixation Test (B. ovis-CFT)

To perform the *B. ovis*-CFT the heterologous antigen from the rough strain *B. ovis* was used, which can detect antibodies induced by *B. canis* infection. Both IgM and IgG antibodies contribute to the reaction. The *B. ovis* antigen for CFT was prepared according to Alton et al. [46]. The protocol for *B. ovis*-CFT was performed in accordance with the Manual of Diagnostic Tests and Vaccines for Terrestrial Animals [34], with minor modifications. Sera were diluted 1:10 in barbital-buffered saline (BBS) pH 7 ± 0.5, and inactivated at 56 °C for 30 min. Together with the World Organization of Animal Health (WOAH)’s positive and negative *B. ovis* standard sera, two rabbit sera were included as additional test controls: a *B. canis* hyperimmune serum as positive control (showing 100% complement fixation at 1:80 dilution), and a negative serum from a *B. canis* naïve rabbit as negative control (showing 100% of hemolysis at 1:10 dilution). All samples and controls were tested as two-fold dilutions starting from 1:10. Samples showing no hemolysis at dilution 1:10 (cut-off) were considered positive. Results were expressed as the highest dilution showing 100% complement fixation (no hemolysis).

### 2.4. Immunofluorescence Antibody Test (IFAT)

The immunofluorescence antibody test (IFAT) was carried out using a commercial kit for the indirect detection of IgGs specific for *B. canis* in dog sera (MegaFLUO BRUCELLA canis, Diagnostik Megacor, Hörbranz, Austria). Analyses were performed following manufacturing instructions and samples were tested at two-fold dilutions starting from 1:40 (cut-off) up to 1:320. Sera showing green fluorescence comparable to the positive control were considered positive. Results were expressed as the highest dilution of serum showing fluorescence.

### 2.5. Statistical Analysis

Data from the Laboratory Information Management System were imported in MS Access^®^ (Microsoft Access 2019, Redmond, Washington, DC, USA), which was used for cleaning and normalizing the dataset. Serological results were analyzed and interpreted taking into account the results of blood culture that represents the gold-standard test to confirm *B. canis* infection. Cut-off, sensitivity (Se), specificity (Sp), and accuracy (Ac) of individual serological methods were calculated by receiver operating characteristic curve (ROC) analysis [47].

The 95% confidence interval (95% CI) with the indication of lower (l.c.l.) and upper (u.c.l.) credibility levels for Se, Sp, and Ac of individual tests were calculated using a Bayesian approach [48] with a beta distribution (*n* + 1; *n* − *s* + 1), where *n* is the total number of tested samples and *s* are the tested positive samples. Finally, a diagnostic protocol was designed considering mSAT and *B. ovis*-CFT in parallel, and IFAT as in series and its performance were evaluated in terms of Se, Sp, Ac, and 95% CI.

## 3. Results

Sera from 61 *B. canis*-infected and 143 non-infected dogs were selected for the study. In all infected animals included in the study, *B. canis* was isolated by culture from EDTA blood samples. All *B. canis* strains isolated from the infected animals included in the current study belong to the same cluster, confirming a single introduction in the infected kennel subject to sampling [25]. Results of positive and negative sera tested using mSAT, *B. ovis*-CFT, and IFAT are shown in Table 1. We verified the performance of these serological tests considering bacterial isolation as the gold-standard test and classifying the sampled population as infected (true positive) or not-infected (true negative). mSAT and *B. ovis*-CFT correctly detected 59 out of 61 (96.7%) sera from infected animals. In only a few cases, positive reactions to the three tests were observed in the uninfected group, suggesting that non-specific reactions may occur in animals not exposed to *B. canis*.

Among the three serological tests evaluated, mSAT showed the lower diagnostic specificity (92.3%) and IFAT showed the highest specificity (99.3%). IFAT also showed higher diagnostic sensitivity (98.3%), while mSAT and *B. ovis*-CFT seem to have similar performances in terms of sensitivity (96.7%). This suggests that IFAT could have better performances compared to mSAT and *B. ovis*-CFT. In two cases, we observed serum samples testing negative for mSAT or *B. ovis*-CFT but positive for IFAT. Moreover, one infected animal, confirmed by culture, tested positive for *B. ovis*-CFT only.

The antibody titers detected in *B. canis*-infected dogs showed wide variability (Figure 2). For mSAT and *B. ovis*-CFT, the most frequent antibody titer (dilution) was 1:160, recorded for 20 samples (32.8%) and 23 samples (37.7%), Figure 2a,b, respectively. Conversely, the most frequent titer observed for IFAT was 1:320, recorded in 51 samples (83.6%), as shown in Figure 2c.

To assess the performance of these serological tests in terms of Se, Sp, Ac, and 95% CI, ROC analyses and credibility level calculation were applied (Table 2). When considering mSAT, ROC analyses indicated the best accuracy (Ac 96.6%) with a cut-off at a dilution of 1:40 (Se 93.4%, Sp 97.9%) (Appendix A). However, the highest Se was recorded considering the cut-off set at dilution of 1:20 (Se 96.7%, Sp 92.3%, Ac 93.6%) (Appendix A). Similarly, ROC analyses of *B. ovis*-CFT data suggested higher accuracy (Ac 98.0%; Se 95.1%, Sp 99.3%) considering a cut-off set at dilution of 1:20 (Appendix A). However, the highest sensitivity was recorded with a cut-off dilution of 1:10 (Se 96.7%, Sp 96.5%, Ac 96.6%) (Appendix A). Since the application of mSAT and *B. ovis*-CFT was intended to screen tests, the cut-offs with the highest sensitivity were selected (1:20 for mSAT and 1:10 for *B. ovis*-CFT). ROC analyses of IFAT showed that cut-off set at dilution of 1:40 was the most accurate (Ac 99.0%, Se 98.4%, Sp 99.3%) (Appendix A).

As a final step, we combined individual tests in order to develop a diagnostic protocol suitable to detect all potentially infected animals during different stages of infection. First, we calculated the combined sensitivity and specificity of mSAT and *B. ovis*-CFT, used in parallel as screening tests (Table 3). The combination of the two tests resulted in maximum Se (100%) but with a decrease of Sp (89.5%) and Ac (92.6%). The subsequent application of IFAT to confirm samples positive to mSAT and/or *B. ovis*-CFT increased protocol specificity (99.3%) maintaining a good sensitivity (98.4%) and the same accuracy (92.6%) (Table 3).

## 4. Discussion

The present study aimed to develop a diagnostic protocol for the serological diagnosis of *B. canis*, a bacterium that causes brucellosis in dogs and wild Canidae and that can be transmissible to humans. The authors evaluated the performance of three individual tests: the microagglutination test (mSAT), the *B. ovis* complement fixation test (*B. ovis*-CFT), and the indirect fluorescent antibody test (IFAT). They assessed the sensitivity and specificity of each test independently and then considered using the tests in parallel and series to determine the optimal combination with the highest sensitivity and specificity. The results demonstrated that, when used in parallel, mSAT and *B. ovis*-CFT identified 100% of infected animals, but with some false positive reactions; this suggested that they could be applied as screening tests. The subsequent application of the IFAT test on mSAT/*B. ovis*-CFT-positive samples increased protocol specificity. The three tests investigated are based on diverse immunological reactions and different antibody isotypes produced in distinct phases of humoral response. The mSAT is an agglutination test that primarily engages IgM antibodies, typically generated in the early stages of the infection. IgMs are pentameric antibodies known to possess higher avidity, but a lower affinity and less specificity compared to other antibody isotypes such as IgGs [45,49]. Previous studies indicated that not only IgMs but also IgGs contribute to *Brucella* agglutination reaction in cattle and humans [45,49]. The *B. ovis*-CFT involves both IgM and IgG antibodies and has complement binding capacity after forming immune complexes. On the other hand, the IFAT detects only IgG antibodies. One possible explanation is that mSAT is based on an agglutination reaction where IgMs play a major role, while both IgMs and IgGs have complement fixation activities. Therefore, the testing of animals at different stages of infection and with different IgM/IgG ratios may explain these differences. However, samples positive for mSAT but negative for *B. ovis*-CFT were also positive for IFAT, confirming the presence of *B. canis*-specific IgGs in the serum. Another possibility is that the antibodies of individual animals reacted differently to the distinct antigen preparations. In fact, mSAT was performed using a *B. canis* whole antigen while CFT was based on a *B. ovis* strain (whole cell). The limited number of samples did not allow further investigation on purified IgMs or IgGs. Literature data are available on individual tests, but no comparative studies have been described for mSAT and *B. ovis*-CFT [39,46,50]. Data recorded using mSAT and *B. ovis*-CFT supported the development of a protocol that considered the use of the two tests in parallel, leading to 100% Se despite the low specificity. The results of the individual tests showed that the mSAT and *B. ovis*-CFT had the same sensitivity but different specificity, with some samples yielding false-positive reactions. These tests were found to be suitable as screening tests due to their high sensitivity, but they had a lower specificity. The IFAT test, which had the highest sensitivity and specificity, was applied to the samples that tested positive in mSAT and/or *B. ovis*-CFT to increase the specificity of the protocol and reduce the likelihood of false-positive results. It is important to note that different antibodies are produced during different phases of the humoral response to *B. canis* infection. IgM antibodies are generated in the early stages, while IgG antibodies are produced later and have higher affinity and specificity. This difference in antibody production could explain the variations in test results, as different animals may have different stages of infection and different IgM/IgG ratios. Additionally, the use of different antigen preparations in the tests may also contribute to differences in test outcomes. Results also showed that infected animals with bacteremia may produce seronegative results. This is in line with previous observations. This study thus highlights the importance of using both direct and indirect diagnostic tests in parallel and repeating testing 6–8 weeks apart in cases of suspected infection with negative initial serological laboratory results. The finding confirmed by this study that infected animals with bacteremia might test seronegative, emphasizes the need for comprehensive testing protocols. While serological tests are easier to perform than culture, the latter is the only test that may lead to a definitive diagnosis of *B. canis* infection. However, all of these tests have their limitations. Blood cultures for isolation require stringent biosafety conditions due to the zoonotic nature of *B. canis*. Serological tests, on the other hand, are less time-consuming and require fewer materials. However, antibodies against *B. canis* may not be detectable in serum until 5–8 weeks after the onset of infection [2]. Furthermore, the antibody titer can vary considerably during the course of infection, and non-specific reactions can occur [50], leading to false positive or doubtful results. Indeed, as reported in our results, some sera from uninfected dogs (resulting in negative blood culture) showed positive reactions to the three tests. Cross-reactivity with other bacterial antigens might also contribute to non-specific results: *Staphylococcus* spp., *Streptococcus* spp., *Pseudomonas aureoginosa,* and *Bordetella bronchiseptica*, all bacteria that may present cell wall antigens that could induce the production of antibodies capable of cross-reacting with *B. canis* [51]. A non-specific reaction might also occur with hemolyzed sera or sera with high lipid content [42]. This highlights, therefore, the importance of proper sera storage after collection, and their rapid analysis, to avoid false-positive or doubtful results. In order to reduce non-specific results, especially with mSAT, the 2-mercaptoethanol (2-ME), already utilized for other serological methods such as the rapid slide agglutination test (2-ME-RSAT), might be used [11,39]. To address the issue of non-specific reactions and improve the accuracy of serological testing, additional confirmatory tests such as the immunoblotting test performed by using secondary anti-dog IgG and IgM antibodies may be developed. However, the lack of an international standard serum or diagnostic tests recognized at an international level poses challenges to the standardization of serological assays for *B. canis*. So far, no guidelines or protocols recognized at international level are available and *B. canis* is not covered in the *Brucella* chapter of the Manual of Diagnostic Tests and Vaccines for Terrestrial Animals. The study acknowledges some limitations, such as the criteria used to identify infected animals. The validation activities described in this work focused on animals with active infections, which facilitated the detection of antibodies by serological tests. However, *B. canis* infection can also manifest as chronic with intermittent bacteremia, leading to limited sensitivity of serological tests in chronically infected animals [52]. The long-term monitoring of infected populations, including chronically infected animals, will provide a better understanding of the performance of the diagnostic protocol over time.

## 5. Conclusions

In most countries, testing for *B. canis* is not mandatory for dog movements and the increasing number of reported infections in dogs, the lack of specific disease surveillance, and the close contacts between humans and dogs contribute to the qualifying of canine brucellosis as an emerging Public Health concern. For a better understanding of the current epidemiology of the disease, reliable laboratory tests are needed.

This study successfully established a diagnostic protocol for *B. canis* using a combination of serological tests. The parallel use of mSAT and *B. ovis*-CFT provided higher sensitivity, while the following application of IFAT improved the specificity of the protocol. The findings emphasize the importance of combining different diagnostic tests, repeating testing over time, and considering the dynamics of the immune response in the diagnosis of *B. canis* infection. This protocol can contribute to a more accurate serological diagnosis of *B. canis* infection in dogs, aiding in the control and management of this emerging public health concern, by improving the current capabilities in the epidemiological investigations on this zoonotic disease [25,42,49,50]. However, further research and standardization efforts are needed to improve the reliability and accuracy of serological testing for *B. canis*-infected dogs, in light of identifying related epidemiological patterns in dogs and humans. Commercial breeding kennels should be regularly checked for causes of abortion and international trade rules should foresee testing for *B. canis* in imported breeding dogs. Guidelines or protocols recognized at an international level should be available and, to this aim, a specific chapter on *B. canis* the Manual of Diagnostic Tests and Vaccines for Terrestrial Animals would be advisable.

## Figures and Tables

**Figure 1 microorganisms-11-02162-f001:**
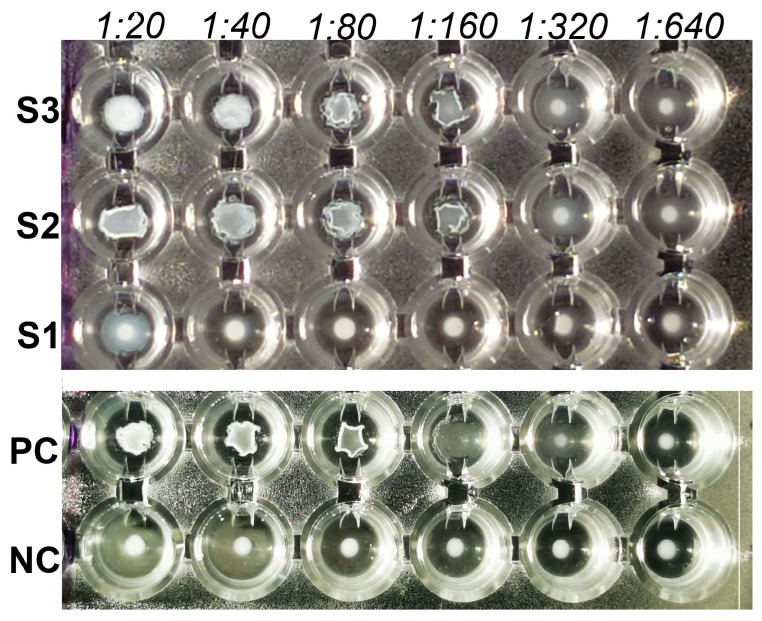
Microplate serum agglutination test (cut-off value < 1:20): positive control (PC, rabbit hyperimmune serum), negative control (NC, negative serum from a *B. canis* naïve rabbit) and 3 dog sera (S1, S2 and S3) tested at dilution from 1:20 to 1:640. PC, S2 and S3 are positive at dilution from 1:20 to 1:160; NC and S1 are negative (no agglutination observed).

**Figure 2 microorganisms-11-02162-f002:**
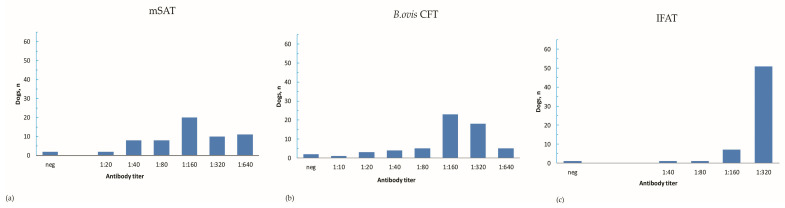
Distribution of antibody titers observed in *B. canis*-infected dogs according to the three serological tests: (**a**) titers observed for mSAT; (**b**) titers observed for *B. ovis*-CFT; (**c**) titers observed for IFAT.

**Table 1 microorganisms-11-02162-t001:** Comparison of microplate serum agglutination test (mSAT), *B. ovis*-complement fixation test (*B. ovis*-CFT), and an immunofluorescence antibody test (IFAT) (Diagnostik Megacor) for the diagnosis of brucellosis, performed on a panel of sera from *B. canis*-infected and uninfected dogs.

	Infected (*n*= 61) ^1^		Uninfected (*n* = 143) ^2^	
	Positive, *n* (%)	Negative, *n* (%)	Positive, *n* (%)	Negative, *n* (%)
Microplate serum agglutination test (mSAT)	59 (96.7)	2 (3.3)	11 (7.7)	132 (92.3)
*B. ovis*-complement fixation test (*B. ovis*-CTF)	59 (96.7)	2 (3.3)	5 (3.5)	138 (96.5)
Immunofluorescence antibody test (IFAT)	60 (98.3)	1 (1.7)	1 (0.7)	142 (99.3)

^1^ Confirmed by *Brucella canis* isolation. ^2^ From other kennels.

**Table 2 microorganisms-11-02162-t002:** Diagnostic accuracy of mSAT, *B. ovis*-CFT, and IFAT was calculated considering a panel of sera from 61 *B. canis*-infected and 143 uninfected dogs and comparing with bacterial isolation of *Brucella canis*.

	mSAT ^2^	*B. ovis*-CTF ^3^	IFAT ^4^
Sensitivity, % (95% CI ^1^)	96.7 (88.8–98.7)	96.7 (88.8–98.7)	98.4 (91.3–99.4)
Specificity, % (95% CI ^1^)	92.3 (86.7–95.1)	96.5 (92.1–98.2)	99.3 (96.2–99.8)
Accuracy, % (95% CI ^1^)	93.6 (89.4–96.2)	96.6 (93.1–98.3)	99.0 (96.5–99.7)

^1^ CI = confidence interval: ^2^ mSAT = Microplate agglutination test. ^3^
*B*. *ovis*-CFT = *B. ovis* Complement fixation test. ^4^ IFAT = Immunofluorescence antibody test.

**Table 3 microorganisms-11-02162-t003:** Sensitivity (Se), specificity (Sp), and accuracy (Ac) of the protocol for serological diagnosis of *B. canis* considering: (i) mSAT and *B. ovis*-CFT in parallel; (ii) IFAT as in series on mSAT and/or *B.ovis*-CFT positive samples. Confidence intervals (95% CI) for Se, Sp, and Ac were calculated with beta distribution.

	mSAT ^2^ + *B. ovis*-CFT ^3^	(mSAT ^2^ *+ B. ovis*-CFT ^3^) + IFAT ^4^
Sensitivity, % (95% CI ^1^)	100.0 (95.3–100.0)	98.4 (91.3–99.6)
Specificity, % (95% CI ^1^)	89.5 (83.4–93.5)	99.3 (96.2–99.8)
Accuracy, % (95% CI ^1^)	92.6 (88.2–95.5)	92.6 (88.2–95.5)

^1^ CI = confidence interval. ^2^ mSAT = Microplate agglutination test. ^3^
*B*. *ovis*-CFT = *B. ovis* Complement fixation test. ^4^ IFAT = Immunofluorescence antibody test.

## Data Availability

The data presented in this study are available upon request from the corresponding author. The data are not publicly available due to privacy.

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
