# Peer review of "Evaluation of Three Serological Tests for Diagnosis of Canine Brucellosis"

_microorganisms, 2023, doi:10.3390/microorganisms11092162_

Round 1

Reviewer 1 Report

In the manuscript entitled, " Evaluation of three Serological Tests for Diagnosis of Canine  Brucellosis Caused by Brucella canis", the authors assessed three serological tests (microplate serum agglutination, complement fixation, and immunofluorescence assay) individually or combined for the diagnosis of B. canis. The manuscript is generally well-addressed and well-written however, I have some comments/edits/suggestions:

Title: need to be revised. It could be “Evaluation of three Serological Tests for Diagnosis of Canine Brucellosis”

Abstract: please revise and add the significance results to abstract. Abstract should provide the readers with the important results.

Line 59: LPS please clarify the abbreviation.

Line 77: "False-positive" F should be lowercase.

Line 97: " Sera were collected from infected animals......" Please specify how many samples were collected from infected and uninfected control animals?

Line 102: The sentence " samples were analysed using Brucella’selective media ...." which type of medium have been used? and which growth conditions being used? Please add them to the text.

Line 104: Please add the PCR conditions being used for positive Brucella spp. colonies screening?

Figure 1: I suggest if possible to include a real positive test results instead of an example of positive results.

Line 144: " WOAH" please clarify abbreviation.

Line 164: what are epidemiological data that were collected? How you used them in the results? I could not see this in results.

In Table 1: The uninfected animals have showed positive results, is that false positive? please discuss in the results/discussion.

Line 191: In the text “ ....correctly identified 60 (98.4%) sera from infected dogs " however, Table 1 ,  IFAT % is 98.3 %. Please correct this and be consistence.

Line 198: respectively (a) (b). it need to be rewrite to be figure 2 (a & b), same with (c)

Line 225 & 229: Table 3? it should be Table 2.

References:

In the text, the reference numbers placed in Italic brackets. Please revise

Reference #23: “Rec. 2017, 180, 384-+”. please correct it.

English quality is fine.

Reviewer 2 Report

Dear Authors

Canine brucellosis, due to its zoonotic potential, poses a significant public health concern. The present study effectively addresses this important issue, making it of great interest to the readers of this journal.

The authors conducted a comprehensive investigation using samples obtained from an outbreak in a commercial kennel. They employed a protocol for serological diagnosis that involved a combination of three different serological tests, namely two in-house laboratory procedures (microplate serum agglutination and complement fixation) and a commercial immunofluorescence test.

I found the main objective of the study highly intriguing, and it is evident that the authors dedicated considerable effort to its execution. This paper holds substantial importance as it has the potential to assist other researchers working in the field. However, I do have some significant concerns regarding the "Results" section that require improvement. Additionally, certain tables need enhancements, and one table appears to be missing. Therefore, I would recommend that publication be considered only after addressing these important issues.

Kind regards.

Suggestions:

Abstract: To address the desire for more results, we will focus on presenting a concise summary of the study's findings rather than extensively discussing the study's rationale and background.

Introduction: Please ensure that a reference is inserted at line 52 to provide appropriate citation and enhance the credibility of the study. Lines 59-60: The current wording is difficult to comprehend. Kindly rephrase and provide additional contextual information to improve clarity. Line 70: This line seems to repeat the information already mentioned in line 57. Please consider revising or removing it to avoid redundancy.

Materials: For Figure 1, I recommend including a schematic diagram adjacent to the image to facilitate better understanding. The addition of a visual aid would enhance the interpretation of the image.

Sampling Size: Could you please clarify how the sampling size was determined? Was it calculated using specific parameters or based on the positive group? Sharing this information would provide valuable context for the study.

Results:

Table 1: To ensure clarity, it is advisable to avoid using acronyms in the table. I suggest combining Table 1 and Table 2 for improved readability and understanding.

Lines 189-191: These lines appear to repeat the information already presented in the table. Consider revising or providing additional insights to avoid redundancy.

Table 3: It seems that Table 3 is missing. Please ensure that it is included to complete the presentation of results.

Reviewer 3 Report

Brucellosis. Among the mSAT, B. ovis-CFT and IFAT methods, IFAT seems to be the best accuracy and dilution. The author makes an indepth discussion from Clinical diagnosis. I clearly recommend the manuscript for publication in Microorganisms.

Minor Comments:

1.    Did the authors consider infections other than Canine Brucellosis,

2.    That method is more accurate at different stages of infection, individual and combined tests, or without knowing the infection

3.    It is suggested that if the sample size is appropriate, it can be used in more sample tests or serum tests of different animals to evaluate the stability of the results
